# Brazilian Green Propolis Rescues Oxidative Stress-Induced Mislocalization of Claudin-1 in Human Keratinocyte-Derived HaCaT Cells

**DOI:** 10.3390/ijms20163869

**Published:** 2019-08-08

**Authors:** Kana Marunaka, Mao Kobayashi, Shokoku Shu, Toshiyuki Matsunaga, Akira Ikari

**Affiliations:** 1Laboratory of Biochemistry, Department of Biopharmaceutical Sciences, Gifu Pharmaceutical University, Gifu 501-1196, Japan; 2Education Center of Green Pharmaceutical Sciences, Gifu Pharmaceutical University, Gifu 502-8585, Japan

**Keywords:** claudin-1, hydrogen peroxide, phosphorylation

## Abstract

Claudin-1 (CLDN1) is expressed in the tight junction (TJ) of the skin granular layer and acts as a physiological barrier for the paracellular transport of ions and nonionic molecules. Ultraviolet (UV) and oxidative stress may disrupt the TJ barrier, but the mechanism of and protective agents against this effect have not been clarified. We found that UVB and hydrogen peroxide (H_2_O_2_) caused the internalization of CLDN1 and increased the paracellular permeability of lucifer yellow, a fluorescent marker, in human keratinocyte-derived HaCaT cells. Therefore, the mechanism of mislocalization of CLDN1 and the protective effect of an ethanol extract of Brazilian green propolis (EBGP) were investigated. The UVB- and H_2_O_2_-induced decreases in CLDN1 localization were rescued by EBGP. H_2_O_2_ decreased the phosphorylation level of CLDN1, which was also rescued by EBGP. Wild-type CLDN1 was distributed in the cytosol after treatment with H_2_O_2_, whereas T191E, its H_2_O_2_-insensitive phosphorylation-mimicking mutant, was localized at the TJ. Both protein kinase C activator and protein phosphatase 2A inhibitor rescued the H_2_O_2_-induced decrease in CLDN1 localization. The tight junctional localization of CLDN1 and paracellular permeability showed a negative correlation. Our results indicate that UVB and H_2_O_2_ could induce the elevation of paracellular permeability mediated by the dephosphorylation and mislocalization of CLDN1 in HaCaT cells, which was rescued by EBGP. EBGP and its components may be useful in preventing the destruction of the TJ barrier through UV and oxidative stress.

## 1. Introduction

The skin forms a barrier between the body and its external environment in order to prevent the intrusion of pathogens and the uncontrolled loss of internal water and solutes. The epidermis is the outermost layer of skin and provides the first line of defense against ultraviolet (UV) radiation and other environmental factors. UV is divided into three ranges based on wavelength: UVA (320–400 nm), UVB (280–320 nm), and UVC (100–290 nm). Among these, UVC is unlikely to be present in terrestrial sunlight because it is blocked by the ozone layer, whereas UVA and UVB can come into contact with the skin. The skin can be separated by the basement membrane into two layers, the dermis and epidermis. The mammalian epidermis is composed of four layers, including the basal, spinous, granular, and cornified layers [1]. UVA can penetrate the dermal layer of the skin and reach the capillaries, whereas UVB is blocked in the upper dermis. According to wavelength-dependent studies, UVB radiation has more cytotoxic and mutagenic effects than UVA does [2,3]. Long-term exposure to UVB induces damage to both the dermal and epidermal skin [4]. UVB-induced cell death is caused through directly induced DNA damage and indirect action mediated by the generation of reactive oxygen species (ROS) and nitric oxide [5]. The main causes of UV-induced DNA damage are 2,3-cyclobutane pyrimidine dimer and a pyrimidine (6-4) pyrimidone photoproduct [6]. In contrast, it is not fully understood what mechanisms are involved in damage to the skin barrier under low toxic conditions. An investigation of the effect of each toxic factor on the tight junction (TJ) barrier may be needed in order to clarify the molecular mechanism.

Human skin keratinocytes form a TJ at the most apical pole of the lateral membrane between neighboring cells. The TJ prevents the abnormal paracellular movement of water, solutes, and pathogens. Claudins (CLDNs) and occludin are integral membrane proteins in TJ, and they comprise a large family of 27 subtypes in mammals [7,8]. Among them, CLDN1 and CLDN4 are highly expressed in human keratinocytes [9]. Most CLDNs contain carboxyl-terminal PSD95/Dlg/ZO-1 (PDZ)-binding motifs that mediate interactions between CLDNs and scaffolding proteins such as ZO-1 and ZO-2. These scaffolding proteins indirectly link CLDNs to actin filaments. CLDN1-deficient mice develop aberrant stratum corneum barrier functions in the skin [10]. A premature stop codon of CLDN1, resulting in a lack of CLDN1 protein, has been identified in neonatal ichthyosis and sclerosing cholangitis syndrome, a disorder characterized by scalp hypotrichosis, ichthyosis, scarring alopecia, and sclerosing cholangitis [11]. Therefore, CLDN1 may have an important role in maintaining the barrier function in the skin.

An antioxidant effect is one of the key factors in reducing skin injury, aging, and cancer risk, and antioxidant activities have been reported in components extracted from plants, fruits, vegetables, and bee propolis [12,13]. Propolis contains many classes of compounds, including flavonoids, phenolic acids, and others. The components of propolis are different in each area [14]. The ethanol extract of Brazilian green propolis (EBGP) exerts strong antioxidant activity in mouse skin [15] and protects human keratinocytes against UV-induced apoptosis [16]. In addition, hydroalcoholic extracts of Brazilian propolis improve dermal burn healing [17]. EBGP may be useful in protecting skin damage from various external stimuli, but the effect of EBGP on the TJ barrier has not been examined.

Human keratinocyte-derived HaCaT cells can easily be plated as a monolayer and form the TJ [18]. Therefore, they may be useful in examining the function of TJ in the skin. In the present study, we investigated the effects of ROS and EBGP on the cellular localization and function of CLDN1 in HaCaT cells. In addition, the molecular mechanism of tight junctional localization of CLDN1 was assessed by immunoprecipitation, immunoblotting, and immunofluorescence measurements.

## 2. Results

### 2.1. Effect of UVB Radiation on the Production of ROS

The production of ROS and H_2_O_2_ in HaCaT cells was measured using 2′,7′-dichlorodihydrofluorescein diacetate (H_2_DCFDA) and Hydrop, respectively. UVB radiation dose-dependently induced the fluorescence intensities of H_2_DCFDA and Hydrop (Figure 1A,B). UVB radiation between 10 and 50 mJ/cm^2^ may have increased the production of ROS and H_2_O_2_. Cell viability was dose-dependently decreased by exposure to UVB (Figure 1C). The effects were significant over 10 mJ/cm^2^, but the percentage of cell toxicity was less than 40%. UVB radiation up to 50 mJ/cm^2^ may have produced relatively low toxic conditions. To clarify the effects of ROS on viability, the cells were transiently exposed to H_2_O_2_ for 3 h. H_2_O_2_ dose-dependently decreased cell viability (Figure 1D). We decided to treat the cells with 50 mJ/ cm^2^ UVB or 200 μM H_2_O_2_ in subsequent experiments.

### 2.2. Effect of H_2_O_2_ on the Cell Localization of TJ Protein

Immunofluorescence measurements showed that CLDN1, CLDN4, and ZO-1 were localized at the TJ under control conditions (Figure 2A,B). The fluorescence signal of CLDN1 at the TJ was not changed by treatment with 200 μM H_2_O_2_ for 3 h, but it became weaker at 6 h. In contrast, those of CLDN4 and ZO-1 were unchanged. CLDN1 was relocalized at the TJ after 48 h. The amount of CLDN1 protein was unchanged by H_2_O_2_ treatment, but that of CLDN4 transiently increased at 6 h (Figure 2C).

### 2.3. Effects of EBGP on the UVB- and H_2_O_2_-Induced Destruction of the TJ Barrier

The function of the TJ barrier was evaluated using transepithelial electrical resistance (TER) and lucifer yellow (LY) flux with HaCaT cells cultured on Transwell inserts. TER decreased after 6 h of treatment with UVB or H_2_O_2_, then returned to the control level after 48 h (Figure 3A). EBGP-induced cell toxicity was examined using a 2-(4-Iodophenyl)-3-(4-nitrophenyl)-5-(2,4-disulfophenyl)-2*H*-tetrazolium (WST-1) assay. Cell viability was not significantly changed by EBGP at concentrations less than 20 μg/mL, but decreased at 50 μg/mL (Figure 3B). The H_2_O_2_-induced mislocalization of CLDN1 was significantly rescued by 10 μg/mL EBGP (Figure 3C), but not by 50 μg/mL EBGP. Similarly, the TJ localization of CLDN1 was decreased by UVB, which was rescued by 10 μg/mL EBGP. Both H_2_O_2_ and UVB caused a decrease in TER and an increase in LY flux, indicating that H_2_O_2_ and UVB induced the destruction of the TJ barrier. These effects were rescued by EBGP (Figure 3D). These results indicated that UVB and H_2_O_2_ may have induced the destruction of the TJ barrier mediated by the loss of CLDN1 in the TJ, which was rescued by EBGP.

### 2.4. Effect of CLDN1 Expression on the TJ Barrier

To confirm the involvement of CLDN1, we examined the effect of knockdown of CLDN1 expression. The mRNA expression of CLDN1 was decreased by the introduction of CLDN1 small interfering RNAs (siRNAs) (Figure 4A). The knockdown of CLDN1 induced a decrease in TER and an increase in LY flux (Figure 4B), indicating that CLDN1 had an important role in the regulation of the TJ barrier. H_2_O_2_ significantly changed neither TER nor LY flux in CLDN1 knockdown cells (Figure 4C). In addition, EBGP showed no recovery effects on the TJ barrier. These results indicated that CLDN1 was necessary for the EBGP-induced rescue effect of the TJ barrier.

### 2.5. Effects of Endocytosis Inhibitors on the Localization of CLDN1

The localization of plasma membrane protein is regulated by endocytotic processes, including clathrin- and caveolae-dependent pathways. The H_2_O_2_-induced mislocalization of CLDN1 was inhibited by monodansylcadaverine (MDC), a clathrin-dependent endocytosis inhibitor, but not by methyl-β-cyclodextrin (MβCD), a caveolae-dependent endocytosis inhibitor (Figure 5A), indicating that H_2_O_2_ may increase the internalization of CLDN1 mediated by the clathrin-dependent endocytosis pathway. An increase in the paracellular permeability caused by H_2_O_2_ was rescued by MDC, but not by MβCD (Figure 5B). These results coincided with the effect of MDC on the H_2_O_2_-induced mislocalization of CLDN1 in the immunofluorescence measurements. The recovery effect of MDC was blocked by the knockdown of CLDN1 expression. These results support that the disappearance of CLDN1 in TJ may induce disruption of the TJ barrier. The localization of some CLDNs has been reported to change according to their phosphorylation status [19,20,21]. Therefore, we decided to investigate the effect of H_2_O_2_ on the phosphorylation of CLDN1.

### 2.6. Effects of Phosphorylation Mimic Mutants of CLDN1 on the TJ Barrier

H_2_O_2_ transiently decreased the phosphothreonine (p-Thr) levels of CLDN1 after 3–6 h (Figure 6A). In contrast, phosphoserine (p-Ser) levels were unchanged by H_2_O_2_ (data not shown). The H_2_O_2_-induced dephosphorylation of CLDN1 was inhibited by EBGP (Figure 6B). It has been reported previously that CLDN1 may be phosphorylated at T191 and T195 [22]. To clarify the dephosphorylation site of CLDN1 associated with H_2_O_2_ treatment, we investigated the effects of phosphorylation mimic mutants of CLDN1, T191E, and T195E. In the absence of H_2_O_2_, the wild-type (WT), T191E, and T195E proteins were localized in the TJ area. H_2_O_2_ increased the intracellular distribution of WT and T195E, but T191E was still mainly localized in the TJ area (Figure 6C). Compared to WT in the absence of H_2_O_2_, T191E showed a similar TJ barrier function, but T195E did not (Figure 6D). These results indicated that the phosphorylation of T191 may be necessary for the TJ localization of CLDN1.

### 2.7. Phosphorylation of CLDN1 by PKC

The phosphorylation level of CLDN1 is regulated by atypical PKC and protein phosphatases (PPs) [23]. Go6976, a selective PKCα and β isoform inhibitor, decreased p-Thr levels and the TJ localization of CLDN1 (Figure 7A,B). H_2_O_2_ decreased PKC activity and increased PP activities, which were significantly inhibited by EBGP (Figure 7C). To support the involvement of PKC, we examined the effects of phorbol 12-myristate 13-acetate (PMA), a PKC activator, and cantharidin, a PP2A inhibitor, on the TJ barrier. The H_2_O_2_-induced mislocalization of CLDN1 was inhibited by both PMA and cantharidin (Figure 7D). Similarly, the reduction in the TJ barrier function by H_2_O_2_ was significantly rescued by both PMA and cantharidin (Figure 7E,F). The recovery effects of PMA and cantharidin were blocked by the knockdown of CLDN1 expression. These results indicated that PKC activator and PP inhibitor could rescue the H_2_O_2_-induced mislocalization of CLDN1.

## 3. Discussion

Disruption of the TJ barrier in the skin is caused by UV exposure and oxidative stress. In previous research, TJ integrity has been examined using high doses of UV, which can induce noticeable cell damage [24]. The production of ROS and H_2_O_2_ was increased by UVB in a dose-dependent manner (Figure 1A,B). ROS production has been reported to be increased by UVB irradiation in a dose- and time-dependent manner [25]. Park et al. [26] reported that ROS production was increased 1.3-fold by UVB irradiation (10 mJ/cm^2^) and incubation for 8 h in HaCaT cells, which is similar to our data. Transient H_2_O_2_ treatment increased the production of ROS and decreased cell viability in a dose-dependent manner (Figure 1D,E). The percentage of damaged cells was about 10%–20% in the present experimental conditions of transient H_2_O_2_ treatment. Transient H_2_O_2_ (200 μM) treatment induced the mislocalization of CLDN1 without affecting the amount of CLDN1 protein (Figure 2). There was no apparent change in CLDN4 localization, but the amount of CLDN4 protein increased transiently after 6 h of H_2_O_2_ treatment. Recently, El-Chami et al. [27] reported that H_2_O_2_ (1 mM, a higher concentration than in our study) induced the mislocalization of CLDN1, CLDN4, and occludin in a continuous cell line of rat epidermal keratinocytes. We suggest that oxidative stress selectively induces the mislocalization of CLDN1 in low-level cell toxic conditions.

The protein level of CLDN1 is decreased in atopic dermatitis, whereas that of CLDN4 is increased [28]. The loss of CLDN1 localization in the TJ may induce a compensatory elevation of CLDN4 expression. Nevertheless, the barrier function of TJ was decreased after 6 h of transient H_2_O_2_ treatment (Figure 3A). The TJ barrier function was reduced by the knockdown of CLDN1 by siRNA (Figure 4A,B). Furthermore, the rescue effects of EBGP on the decrease in TER and increase in LY flux by H_2_O_2_ were blocked by the knockdown of CLDN1 (Figure 4C). There results indicate that CLDN1 has an important role in the maintenance of the TJ barrier. Previously, Furuse et al. [29] have reported that a continuous TJ was observed in the stratum granulosum of the epidermis in CLDN1-deficient mice and in WT mice using ultrathin section electron microscopy. However, a small molecule tracer (~600 D) passes through the TJ of the epidermis in CLDN1-deficient mice, whereas the diffusion is prevented at the TJ in WT mice. We suggest that CLDN1 cannot be replaced by CLDN4 in the skin.

The H_2_O_2_-induced mislocalization of CLDN1 was inhibited by a clathrin-dependent endocytosis inhibitor (Figure 5A), suggesting that H_2_O_2_ enhances the endocytosis of CLDN1 from the TJ to intracellular compartments. Although the protein levels of CLDN1 did not change, we did not detect the subcellular localization of CLDN1 in the organelle (Figure 2). We have to clarify the subcellular localization of CLDN1 using organelle markers in further studies. The p-Thr level of CLDN1 was decreased by H_2_O_2_ (Figure 6A). The phosphorylation status of proteins is regulated by various protein kinases and PPs. Go6976 induced the dephosphorylation and mislocalization of CLDN1 (Figure 7). On the contrary, the H_2_O_2_-induced mislocalization of CLDN1 was rescued by the PKC activator PMA. These results suggest that the TJ localization of CLDN1 is upregulated by PKC-dependent phosphorylation. Dephosphorylation of CLDN1 is caused by the activation of PP2A [23]. H_2_O_2_ increased PP activities, and the mislocalization of CLDN1 was rescued by a PP inhibitor, cantharidin, indicating that PPs may also be involved in the dephosphorylation of CLDN1 by H_2_O_2_. CLDN1 is phosphorylated in both renal tubular MDCK I and colonic HT29 cells, but PKC-induced changes in the phosphorylation state were detected only in MDCK I cells [21]. The regulatory mechanism for the phosphorylation of CLDN1 may be different in each tissue.

Threonine phosphorylation sites of CLDN1 by PKC were predicted at T191 and T195 using the NetPhos 2.0 and Disphos 1.3 servers [22]. A phosphorylation mimic T191E mutant was localized to the TJ and maintained the TJ barrier in cells treated with H_2_O_2_, whereas WT and a T195E mutant lost their localization and barrier function in the TJ (Figure 6C,D). These results indicate that phosphorylation at T191 may be necessary for CLDN1 to localize to the TJ in HaCaT cells. The necessity of phosphorylation at T191 has been reported using human embryonic kidney 293 cells [30] and MDCK cells [31].

H_2_O_2_ caused a decrease in CLDN1 localization in TJ and an increase in paracellular permeability, which were rescued by EBGP (Figure 3C,D). Similar effects were observed in the UVB-treated cells. Although the effect of EBGP may be due to antioxidant activity, some cinnamic acid derivatives and flavonoids, which are contained in propolis, have beneficial effects on PKC. Artepillin C enhances adipocyte differentiation and glucose uptake mediated by the activation of PKC [32]. Chlorogenic acid prevents α-amino-hydroxy-5-methyl-isoxazole-4-propionate-mediated excitotoxicity in optic nerve oligodendrocytes through a PKC-dependent pathway [33]. Caffeic acid phenethyl ester inhibits the expression and activity of PP2A [34]. Our preliminary data indicate that kaempferide, which is abundant in EBGP [35], had lower antioxidant activity compared to EBGP, but it rescued the H_2_O_2_-induced mislocalization of CLDN1 and the reduction in the TJ barrier function. The components of propolis, including kaempferide, vary depending on the area [14]. A comparison of the effects of propolis produced in various places may be good for the identification of active components. Further studies are needed on which components of EBGP can rescue the H_2_O_2_-induced mislocalization of CLDN1 using human keratinocytes and what doses are effective. The identification of active components could lead to expanding the range of raw substances beyond green propolis, which could prove useful for the same functions.

In conclusion, we found that UVB irradiation increased ROS production, including H_2_O_2_, and both UVB and H_2_O_2_ caused the mislocalization of CLDN1 in HaCaT cells. The H_2_O_2_-induced mislocalization of CLDN1 was rescued by EBGP, PMA, and cantharidin. H_2_O_2_ decreased PKC activity and increased PP activities, which were inhibited by EBGP. Go6976 decreased p-Thr levels and the TJ localization of CLDN1. These results suggest that the TJ localization of CLDN1 is controlled by PKC. H_2_O_2_ decreased the p-Thr level of CLDN1, but this effect was blocked by EBGP. The T191E CLDN1 mutant was localized to the TJ after treatment with H_2_O_2_. We suggest that the phosphorylation of CLDN1 at T191 is necessary for its localization in the TJ. EBGP and its components may be useful in preventing the destruction of the TJ barrier by UVB and oxidative stress.

## 4. Materials and Methods

### 4.1. Materials

Rabbit anti-CLDN1, mouse anti-CLDN4, and rabbit anti-ZO-1 antibodies were obtained from Thermo Fisher Scientific (San Diego, CA, USA). Goat anti-β-actin and mouse anti-FLAG antibodies were from Santa Cruz Biotechnology (Santa Cruz, CA, USA) and Wako Pure Chemical (Osaka, Japan), respectively. Mouse anti-p-Ser and anti-p-Thr antibodies were from Sigma-Aldrich (Saint Louis, MO, USA). EBGP, LY, H_2_DCFDA, and PMA were from Yamada Bee Company, Inc. (Lot: LY-009, Okayama, Japan), Biotium (Fremont, CA, USA), Thermo Fisher Scientific, and LC Laboratories (Woburn, MA, USA), respectively. OxiOrange and Hydrop were from Goryo Chemical (Hokkaido, Japan). All other reagents were of the highest grade of purity available.

### 4.2. Cell Culture

HaCaT cells, an immortalized nontumorigenic human keratinocyte-derived cell line [36], were grown in Dulbecco’s Modified Eagle’s Medium (Sigma-Aldrich) supplemented with 5% fetal bovine serum (FBS, Sigma-Aldrich), 0.07 mg/mL penicillin-G potassium, and 0.14 mg/mL streptomycin sulfate in a 5% CO_2_ atmosphere at 37 °C. One day before experiments, cells were transferred to FBS-free medium. Cell viability was examined using a WST-1 assay.

### 4.3. UVB Irradiation

UVB irradiation was carried out using a UV Crosslinker CL-1000M (Analytik Jena, Upland, CA, USA), which emits most of its energy within the UVB range (peaking at 302 nm). HaCaT cells were irradiated at a dose of 5–50 mJ/cm^2^ in Hank’s balanced salt solution. After UVB radiation, cells were incubated in fresh medium until analysis.

### 4.4. Production of Reactive Oxygen Species

H_2_DCFDA can detect several ROS, including H_2_O_2_, ∙OH, and peroxy radical, whereas OxiOrange and Hydrop selectively detect ∙OH and H_2_O_2_, respectively. HaCaT cells were incubated with these ROS-sensitive fluorescent probes for 30 min. The fluorescence intensity of each probe was detected using an Infinite F200 Pro microplate reader (Tecan, Mannedorf, Switzerland).

### 4.5. Confocal Microscopy

Cells were plated on cover glasses. After forming a confluent monolayer, the cells were incubated with cold methanol for 10 min at −30 °C and then permeabilized with 0.2% Triton X-100 for 10 min. Following permeabilization, the cells were blocked with 4% Block Ace (Dainippon Sumitomo Pharma, Osaka, Japan) for 30 min and incubated with anti-CLDN1, anti-CLDN4, anti-FLAG, or anti-ZO-1 antibodies (1:100 dilution) for 16 h at 4 °C, followed by incubation with Alexa Fluor 488- or 555-conjugated secondary antibodies (1:100 dilution). The fluorescence images were observed using an LSM 700 confocal microscope (Carl Zeiss, Jena, Germany).

### 4.6. SDS-Polyacrylamide Gel Electrophoresis and Immunoblotting

Cells were scraped into cold phosphate-buffered saline and precipitated by centrifugation. They were lysed in a RIPA buffer containing 150 mM NaCl, 0.5 mM EDTA, 1% Triton X-100, 0.1% SDS, 50 mM Tris-HCl (pH 8.0), and a protease inhibitor cocktail (Sigma-Aldrich) and were sonicated for 20 s. After centrifugation at 6000× *g* for 5 min, the supernatants were collected and used as cell lysates, which included membrane and cytoplasmic proteins. Samples were applied to SDS-polyacrylamide gel electrophoresis (SDS-PAGE) and blotted onto a polyvinylidene fluoride membrane. The membrane was then incubated with the respective primary antibody (1:1000 dilution) at 4 °C for 16 h, followed by a peroxidase-conjugated secondary antibody (1:3000 dilution) at room temperature for 1.5 h. Finally, the blots were incubated in EzWestLumi Plus (Atto, Tokyo, Japan) or ImmunoStar Basic (Wako Pure Chemical) and scanned with a C-DiGit Blot Scanner (LI-COR Biotechnology, Lincoln, NE, USA). The blots were stripped and reprobed with an anti-β-actin antibody. Band density was quantified using ImageJ software (National Institute of Health software). The signals were normalized using a β-actin loading control.

### 4.7. Measurement of Paracellular Permeability

Cells were plated on Transwell plates (0.4 μm pore size, Corning Inc., Corning, NY, USA). After forming a confluent monolayer, TER was measured using a Millicell-ERS epithelial volt-ohmmeter (Millipore, Billerica, MA, USA). TER values (ohms × cm^2^) were normalized by the area of the monolayer and were calculated by subtracting the blank values from the filter and the bathing medium. The paracellular permeability to LY, a fluorescent paracellular flux marker, for 1 h from upper to lower compartments was measured with an Infinite F200 Pro microplate reader.

### 4.8. Isolation of Total RNA and Quantitative Real-Time Polymerase Chain Reaction

Total RNA was extracted using a TRI reagent (Sigma-Aldrich). Reverse transcription and quantitative real-time polymerase chain reaction (PCR) was performed as described previously [37]. The primer pairs used for PCR were human CLDN1 (sense: 5′-ATGAGGATGGCTGTCATTGG-3′; antisense: 5′-ATTGACTGGGGTCATAGGGT-3′) and human β-actin (sense: 5′-CCTGAGGCACTCTTCCAGCCTT-3′; antisense: 5′-TGCGGATGTCCACGTCACACTTC-3′).

### 4.9. PKC and Serine/Threonine Protein Phosphatase Activity Assays

Cells were harvested in 1× Passive Buffer (Promega, Madison, WI, USA). PKC activity was measured using a CycLex PKC Super Family Kinase Assay Kit (Medical & Biological Laboratories, Nagoya, Japan) in accordance with the manufacturer’s protocol. Serine/threonine PP activities were investigated using paranitrophenylphosphate (pNPP) as a substrate at pH 7.5 (neutral condition) and pH 8.4 (alkaline condition). The hydrolysis of pNPP was measured by monitoring the absorbance at 405 nm.

### 4.10. Vector Construction and Transfection

A vector containing human CLDN1 cDNA was prepared as described previously [31] and was called CLDN1/pCMV, which encoded a FLAG tag at the amino terminus. Mutants of T191E and T195E were generated as described previously [31]: siRNAs against CLDN1 (SASI_Hs01_00211216) and a negative control (SIC-001) were purchased from Sigma-Aldrich. Plasmid vector and siRNAs were transfected into cells using Lipofectamine 2000, as recommended by the manufacturer.

### 4.11. Statistics

Results are presented as means ± S.E.M. Differences between groups were analyzed using one-way analysis of variance, and corrections for multiple comparison were made using Tukey’s multiple comparison test. Comparisons between two groups were made using Student’s *t*-test. Statistical analyses were performed using KaleidaGraph version 4.5.1 software (Synergy Software, Reading, PA, USA). Significant differences were assumed at *p* < 0.05.

## Figures and Tables

**Figure 1 ijms-20-03869-f001:**
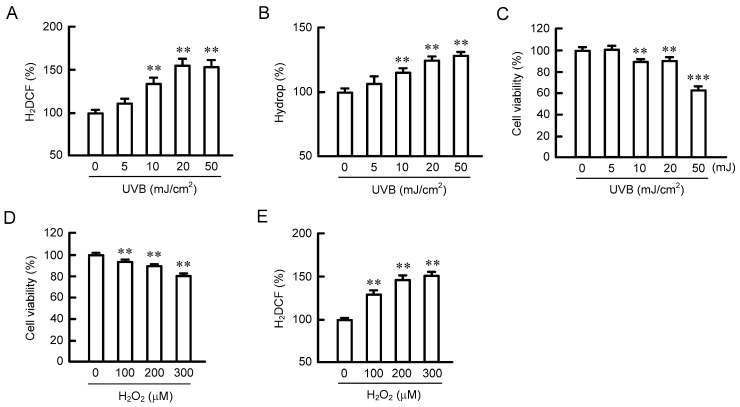
Effects of UVB and H_2_O_2_ on reactive oxygen species (ROS) production and cell toxicity in HaCaT cells. (**A**,**B**) Cells exposed to UVB were cultured for 3 h and then incubated with 2′,7′-dichlorodihydrofluorescein diacetate (H_2_DCFDA) (ROS) and Hydrop (H_2_O_2_) for 30 min. The florescence intensities of each dye were measured using an Infinite F200 Pro microplate reader and are shown relative to the values (0 mJ/cm^2^). (**C**,**D**) Cells were exposed to UVB or transiently exposed to H_2_O_2_ for 3 h, and then they were cultured for 24 h. Cell viability was measured using a WST-1 assay. (**E**) Cells were exposed to H_2_O_2_ for 3 h, and then they were incubated with H_2_DCFDA. The florescence intensities of H_2_DCF are shown as relative to the values in 0 μM H_2_O_2_. *n* = 4–6; *** *p* < 0.001 and ** *p* < 0.01 significantly different from 0 mJ/cm^2^ UVB or 0 μM H_2_O_2_.

**Figure 2 ijms-20-03869-f002:**
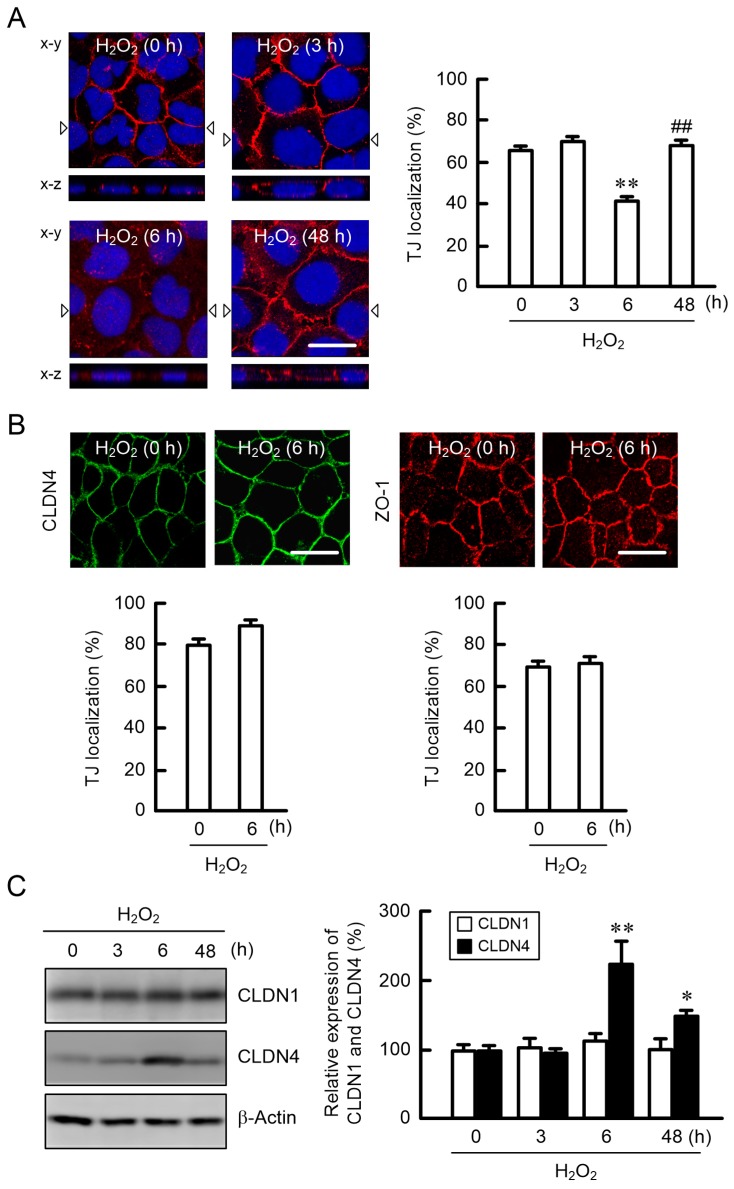
Effects of H_2_O_2_ on the localization and expression of claudins (CLDNs). Cells exposed to 200 μM H_2_O_2_ were cultured for 0, 3, 6, and 48 h. (**A**) The cells were immunostained with anti-CLDN1 (red) antibodies and DAPI (nuclear marker). The low panels (x-z) show the vertical sections indicated by the triangles of x-y images. Scale bars indicate 10 μm. The fluorescence values of CLDN1 at the tight junction (TJ) are shown as a percentage of the total fluorescence values. (**B**) The cells were immunostained with anti-CLDN4 (green) or anti-ZO-1 (red) antibodies. The fluorescence values of CLDN4 and ZO-1 at the TJ are shown as a percentage of the total fluorescence values. (**C**) Cell lysates were applied to 12.5% SDS-PAGE and blotted with anti-CLDN1, anti-CLDN4, and β-actin antibodies. The expression levels of CLDN1 and CLDN4 are represented relative to the values at 0 h. *n* = 3–8; ** *p* < 0.01 and * *p* < 0.05 significantly different from 0 h; ^##^
*p* < 0.01 significantly different from 6 h.

**Figure 3 ijms-20-03869-f003:**
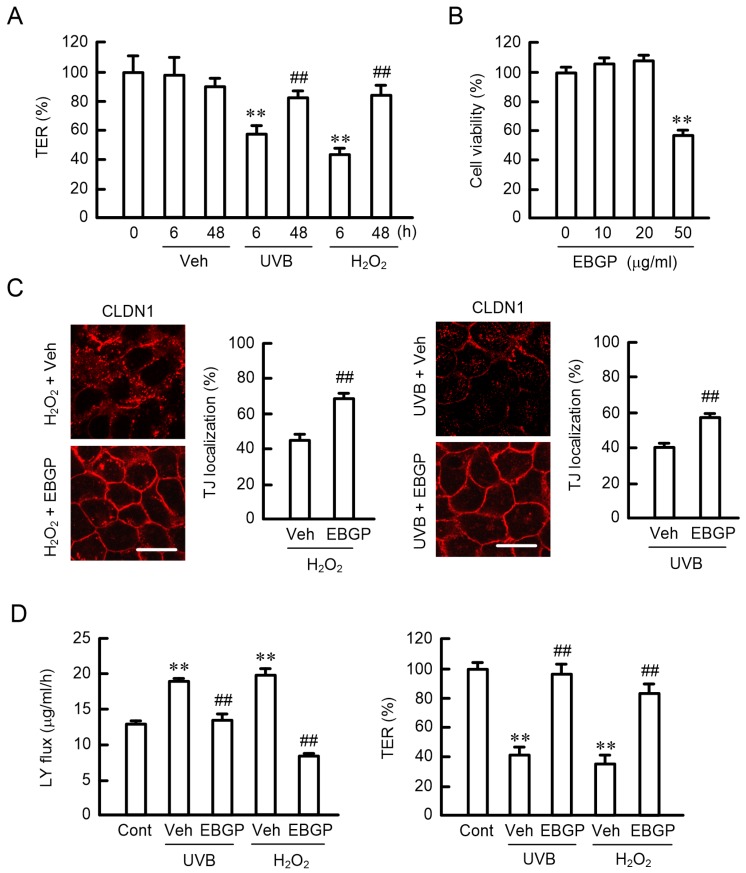
Effects of UVB, H_2_O_2_, and Brazilian green propolis (EBGP) on TJ permeability. (**A**) Cells plated on Transwell inserts were incubated with 50 mJ/cm^2^ UVB or 200 μM H_2_O_2_ for the periods indicated. Transepithelial electrical resistance (TER) was measured using a volt-ohmmeter and is represented as a percentage of the values at 0 h. (**B**) Cells were incubated in the presence or absence of EBGP for 24 h. Cell viability was measured using a WST-1 assay. (**C**) The cells were pre-incubated in the absence (Veh) or presence of 10 μg/mL EBGP for 30 min, followed by exposure to UVB or H_2_O_2_ for 6 h. The cells were immunostained with anti-CLDN1 antibodies. Scale bars indicate 10 μm. The TJ localization of CLDN1 is represented as the percentage of total amount. (**D**) The cells were pre-incubated in the absence (Veh) or presence of 10 μg/mL EBGP for 30 min, followed by exposure to UVB or H_2_O_2_ for 6 h. Control cells (Cont) were not treated with UVB or H_2_O_2_. TER was measured using a volt-ohmmeter and is represented as a percentage of the values in control cells. Paracellular lucifer yellow (LY) flux was analyzed using fluorescence spectrometry. *n* = 4; ** *p* < 0.01 significantly different from 0 h, 0 μg/mL, or Cont; ^##^
*p* < 0.01 significantly different from 6 h of UVB and H_2_O_2_ or Veh of UVB and H_2_O_2_.

**Figure 4 ijms-20-03869-f004:**
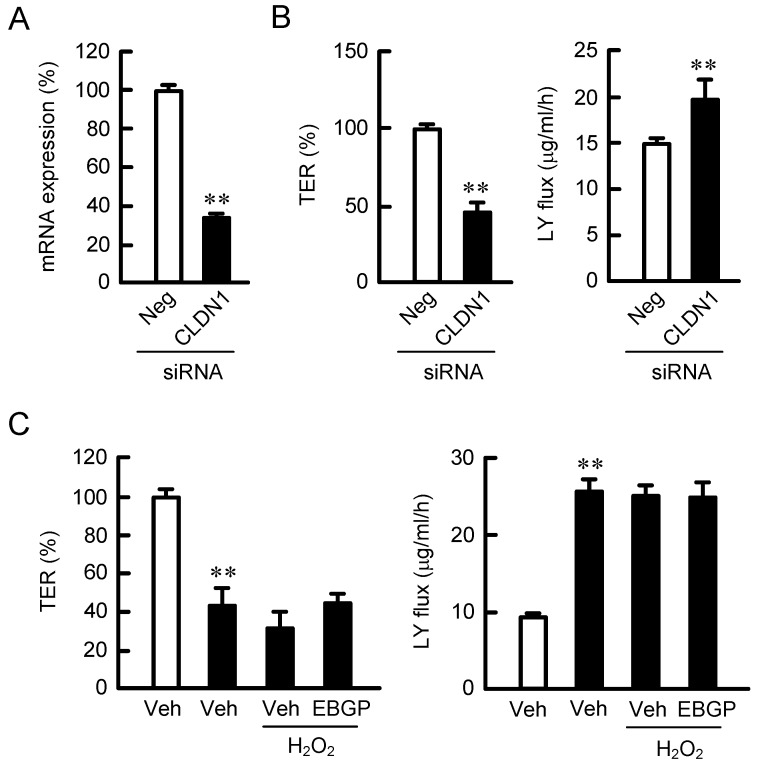
Destruction of the TJ barrier by a decrease in CLDN1 expression. The cells were transfected with negative (Neg and open bars) or CLDN1 (closed bars) small interfering RNAs (siRNAs). (**A**) The expression level of CLDN1 mRNA was measured using real-time PCR. (**B**) TER and paracellular LY flux were analyzed using a volt-ohmmeter and fluorescence spectrometry, respectively. (**C**) The cells were incubated in the absence (Veh) or presence of 200 μM H_2_O_2_ and 10 μg/mL EBGP for 6 h. TER and paracellular LY flux were analyzed. *n* = 4; ** *p* < 0.01 significantly different from Veh.

**Figure 5 ijms-20-03869-f005:**
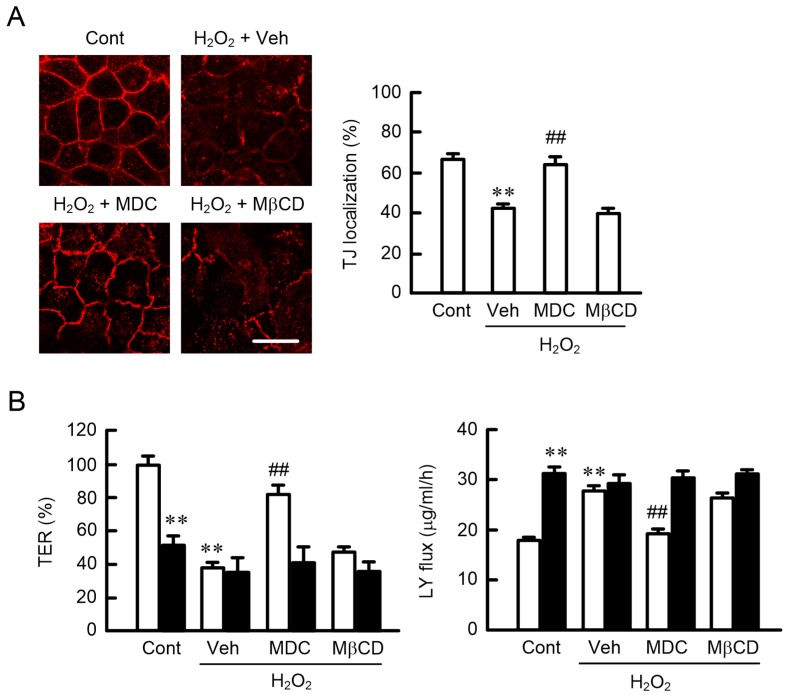
Rescue of the H_2_O_2_-induced mislocalization of CLDN1 by monodansylcadaverine (MDC). (**A**) Cells were pre-incubated in the absence (Veh) or presence of 5 μM MDC and 10 μM methyl-β-cyclodextrin (MβCD) for 30 min, followed by exposure to 200 μM H_2_O_2_. Control cells (Cont) were not treated with H_2_O_2_ and endocytosis inhibitors. The cells were immunostained with anti-CLDN1 antibodies. Scale bars indicate 10 μm. The TJ localization of CLDN1 is represented as the percentage of the total amount. (**B**) Negative (open bars) or CLDN1 siRNA-transfected cells (closed bars) were incubated with H_2_O_2_, MDC, and MβCD. TER and paracellular LY flux were analyzed using a volt-ohmmeter and fluorescence spectrometry, respectively. *n* = 4–6; ** *p* < 0.01 significantly different from Cont or Cont of negative siRNA; ^##^
*p* < 0.01 significantly different from Veh of H_2_O_2_.

**Figure 6 ijms-20-03869-f006:**
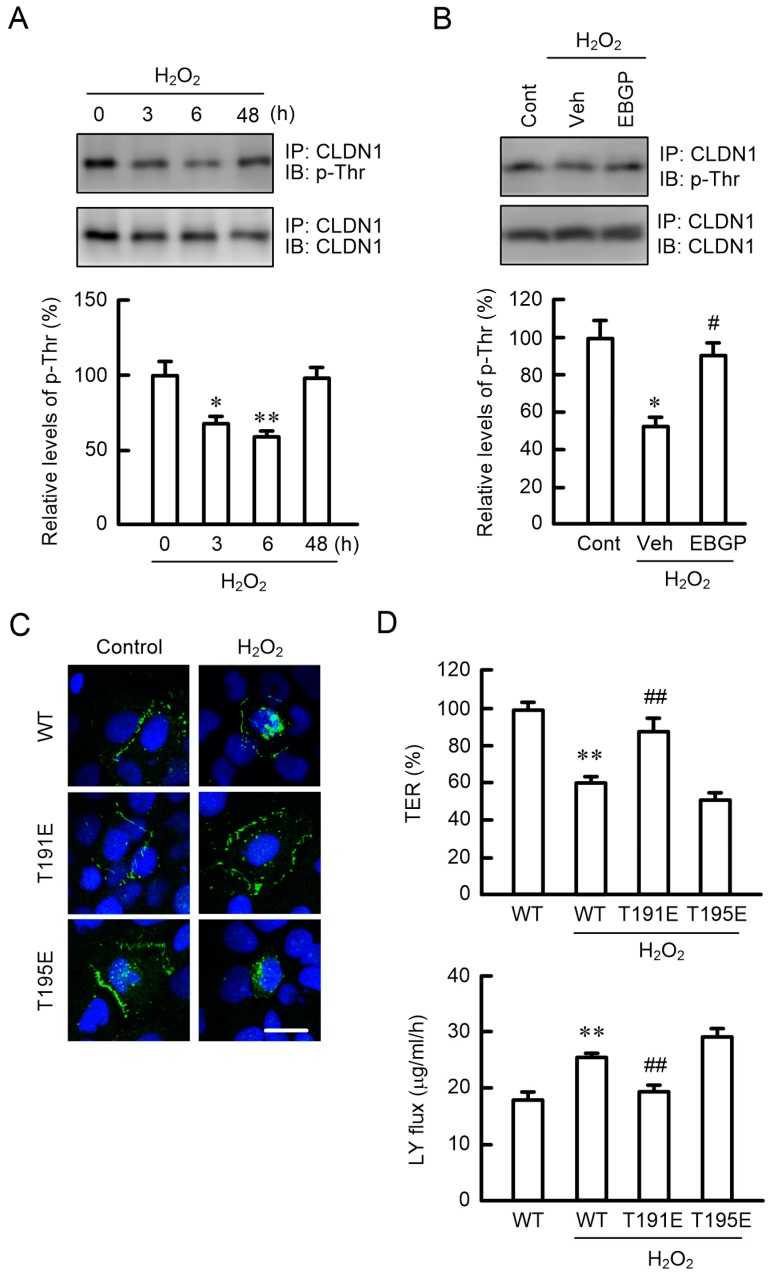
Effects of phosphorylation mimic mutants on H_2_O_2_-induced barrier destruction. (**A**) Cell lysates were prepared from the cells exposed to 200 μM H_2_O_2_ for the periods indicated. (**B**) Cell lysates were prepared from the cells exposed to 200 μM H_2_O_2_ in the absence (Veh) or presence of 10 μg/mL EBGP for 6 h. After immunoprecipitation with anti-CLDN1 antibodies, the immunoprecipitants were applied to 12.5% SDS-PAGE and blotted with antiphosphothreonine (p-Thr) and anti-CLDN1 antibodies. The levels of p-Thr are represented relative to the values at 0 h or the control (Cont). (**C**,**D**) Cells were transfected with FLAG-tagged wild-type (WT), T191E, and T195E CLDN1. At 48 h after transfection, the cells were exposed to 200 μM H_2_O_2_ for 6 h. (**C**) The cells were immunostained with anti-FLAG antibodies (green) and DAPI (blue), a nuclear marker. Scale bars indicate 10 μm. (**D**) TER and paracellular LY flux were analyzed using a volt-ohmmeter and fluorescence spectrometry, respectively. *n* = 3–4; ** *p* < 0.01 and * *p* < 0.05 significantly different from 0 h, Cont, or WT alone; ^##^
*p* < 0.01 and ^#^
*p* < 0.05 significantly different from Veh with H_2_O_2_ or WT with H_2_O_2_.

**Figure 7 ijms-20-03869-f007:**
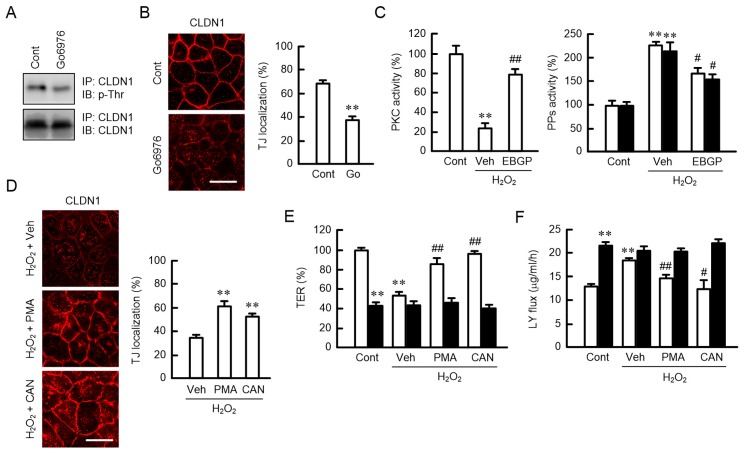
Upregulation of the TJ localization and phosphorylation of CLDN1 by PKC. (**A**,**B**) Cells were incubated in the absence (Cont) or presence of 10 μM Go6976 (Go) for 30 min (**A**) or 6 h (**B**). (**A**) After immunoprecipitation with anti-CLDN1 antibodies, the immunoprecipitants were blotted with anti-p-Thr and anti-CLDN1 antibodies. (**B**) The cells were immunostained with anti-CLDN1 antibodies. Scale bars indicate 10 μm. The TJ localization of CLDN1 is represented as the percentage of the total amount. (**C**) Cell lysates were prepared from the cells exposed to 200 μM H_2_O_2_ in the presence (EBGP) and absence (Veh) of 10 μg/mL EBGP. Control cells (Cont) were not treated with H_2_O_2_ and EBGP. PKC activity was measured using a CycLex PKC Super Family Kinase Assay Kit. Protein phosphatase (PP) activities were measured using pNPP in neutral (open bars) and alkaline (closed bars) conditions. (**D**) The cells were pre-incubated with vehicle (Veh), 100 nM phorbol 12-myristate 13-acetate (PMA), or 1 μM cantharidin (CAN) for 30 min, followed by exposure to 200 μM H_2_O_2_ for 6 h. The cells were immunostained with anti-CLDN1 antibodies. The TJ localization of CLDN1 is represented as the percentage of the total amount. Scale bars indicate 10 μm. (**E**,**F**) Negative (open bars) or CLDN1 siRNA-transfected cells (closed bars) were incubated with H_2_O_2_, PMA, and CAN for 6 h. Control cells (Cont) were not treated with H_2_O_2_ and these drugs. TER and paracellular LY flux were analyzed using a volt-ohmmeter and fluorescence spectrometry, respectively. *n* = 3–8; ** *p* < 0.01 and * *p* < 0.05 significantly different from Cont, Cont of negative, or Veh with H_2_O_2_; ^##^
*p* < 0.01 and ^#^
*p* < 0.05 significantly different from Veh with H_2_O_2_.

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
