# Peer review of "Brazilian Green Propolis Rescues Oxidative Stress-Induced Mislocalization of Claudin-1 in Human Keratinocyte-Derived HaCaT Cells"

_ijms, 2019, doi:10.3390/ijms20163869_

Round 1

Reviewer 1 Report

This is a very interesting paper giving insight into tight junction localisation in HaCaT cells, in relation to UVB and H2O2 exposure. The results are relevant to tight junction barrier function and the figures are well presented in the main and it is well written.

I have a few specific comments:

Page 7, first paragraph: Please define 'toxicity'. Do you mean photo toxicity? Cell toxicity? Be specific.

Page 9: The sentence 'The results coincide with those in immunoflourescence measurements' - this sentence should be revised - it is not clear.

Page 11, last paragraph: again define 'toxic'

Page 27 to 36: Figures 1 to 7: Not necessary to put NS on non-significant bars - remove.

Author Response

We thank you very much for your careful reading of our manuscript and valuable comments.

Comment 1

Page 7, first paragraph: Please define 'toxicity'. Do you mean photo toxicity? Cell toxicity? Be specific.

Answer

  Following your suggestion, we modified it.

Comment 2

Page 9: The sentence 'The results coincide with those in immunoflourescence measurements' - this sentence should be revised - it is not clear.

Answer

  Following your suggestion, we modified the sentence.

Comment 3

Page 11, last paragraph: again define 'toxic'

Answer

  Following your suggestion, we modified it.

Comment 4

Page 27 to 36: Figures 1 to 7: Not necessary to put NS on non-significant bars - remove.

Answer

  Following your suggestion, we deleted NS.

Reviewer 2 Report

In this study, the authors investigated the effects of Brazilian green propolis (EBGP) on oxidative stress-induced mislocalization of claudin-1 and reduced TER in human keratinocyte-derived HaCaT cells. They found that UV and H2O2 led to internalization of claudin 1 and increased paracellular permeability in HaCaT cells. The UVB- and H2O2-induced decrease in claudin 1 localization were rescued by EBGP. H2O2 decreased the phosphorylation of claudin 1 was also rescued by EBGP. Furthermore, both protein kinase C activator and protein phosphatase 2A inhibitor rescued the H2O2-induced decrease in claudin 1 localization. The results suggested that EBGP and its components may be useful to prevent damages of tight junction barrier by UV and oxidative stress.

This is a well-designed study and manuscript is well-written. One minor suggestion is to add a reference to "Human keratinocyte-derived HaCaT cells construct TJ, but do not form multilayer structures" on page 6. 

Author Response

We thank you very much for your careful reading of our manuscript and valuable comments.

Comment 1

  This is a well-designed study and manuscript is well-written. One minor suggestion is to add a reference to "Human keratinocyte-derived HaCaT cells construct TJ, but do not form multilayer structures" on page 6.

Answer

  Thank you very much for your indications. We modified the sentence and added a new reference.